# Surgeon interpretation of patient-reported outcome measures in upper extremity osteoarthritis

Rick S. Looman[1,2], Koen D. Oude Nijhuis[1,2,3], Sina Ramtin[1], David Ring[1]*, Daniel A. London[4], Ryan Calfee[5], Niels Brinkman[1,2,3], Science of variation group¶

1 Department of Surgery and Perioperative Care, Dell Medical School—The University of Texas at Austin, Austin, Texas, United States of America, 2 Department of Orthopaedic Surgery, University Medical Centre Groningen and Groningen University, Groningen, the Netherlands, 3 Department of Trauma Surgery, University Medical Centre Groningen and Groningen University, Groningen, the Netherlands, 4 Department of Orthopedic Surgery, University of Missouri, Colombia, Missouri, United States of America, 5 Department of Orthopaedic Surgery, Washington University School of Medicine, St Louis, Missouri, United States of America

¶ Members from the Science of variation group who participated in responding to the survey are listed in the acknowledgment section.
* David.Ring@austin.utexas.edu

## Abstract

### Purpose

Measures that quantify levels of pain intensity, incapability, symptoms of distress, and unhelpful thinking were designed as research tools. We studied how surgeons interpret and act on these scores in the care of individual patients.

### Methods

In an online experiment, 67 upper limb specialists from the Science of Variation Group reviewed 10 fictious patient scenarios of trapeziometacarpal, wrist and elbow osteoarthritis (OA) and randomized elements of radiographic severity, levels of discomfort and incapability, deprivation index score (social health), and levels of distress and unhelpful thinking (mental health). Multilevel mixed-effects linear regression identified patient factors associated with surgeon concern, enthusiasm to offer surgery, motivation to discuss mental health and social circumstances, and likelihood to refer for mental/social health care.

### Results

Higher levels of discomfort, incapability, distress, unhelpful thinking, and deprivation were associated with greater surgeon concern, increased motivation to discuss or refer for mental and social health, and greater motivation to refer for mental/social health. Greater likelihood to offer surgery was associated with greater discomfort and

**Data availability statement:** All relevant data are within the paper and its Supporting Information files.

**Funding:** The author(s) received no specific funding for this work.

**Competing interests:** I have read the journal's policy and the authors of this manuscript have the following competing interests: David Ring reports other from Skeletal Dynamics, personal fees from Deputy Editor for Clinical Orthopaedics and Related Research, personal fees from Universities and Hospitals, personal fees from Lawyers, personal fees from Health Services and Resource Administration and Department of Justice, personal fees from Premier Healthcare Solutions, personal fees from Wolters Kluwer Health, grants from National Institutes for Health, personal fees from Everus, other from MyMedicalHub, outside the submitted work. Ryan Calfee receives grants from the NIMH, serves as Deputy Editor in Chief Journal of Hand Surgery and servers as Research Director American Society for Surgery of the Hand. All other authors have no conflicts of interest to declare. [DR] is affiliated with Premier Healthcare Solutions, Wolters Kluwer Health, and Everus. This does not alter our adherence to PLOS ONE policies on sharing data and materials.

incapability, greater radiographic severity, wrist and elbow OA, and greater distress/unhelpful thinking.

## Conclusions

Measures that quantify the subjective aspects of illness draw surgeon attention to mental and social health, and can be used to inform comprehensive, biopsychosocial diagnostic and treatment strategies.

---

## Introduction

### Background

Patient-reported outcome measures (PROMs) quantify health from the patient's perspective. PROMs relevant to musculoskeletal conditions generally quantify levels of comfort (pain intensity) and capability. PROMs were initially developed as research tools and many are lengthy and burdensome. Short-form PROMs and PROMs utilizing computer adaptive testing (CATs) have decreased the burden of measure completion, making them more appealing for routine use. PROMs are incorporated in the electronic medical records at many institutions, offering a potential point of discussion between patient and clinician [1]. There is hope that quantification of musculoskeletal comfort and capability can help align test and treatment options to what matters most to individual patients. The concept is that competition centered on measures of outcomes that are important to patients has the potential to improve health while reducing the variability and cost of care [2,3].

### Rationale

In the research setting, PROMs can help compare the outcomes of two or more test or treatment strategies. In the care of an individual patient, the interpretation of PROM scores may be unclear. PROM scores do not have disease or health benchmarks like a blood pressure or hemoglobin A1C, and a person can have low levels of discomfort and incapability in the presence of severe disease [4–7]. Minimal clinically important difference can be used to evaluate change scores, but are not useful at the initiation of care and have other limitations [8]. There is, to our knowledge, little evidence or consensus among surgeons regarding how measures of the subjective are interpreted and whether surgeons use thresholds to interpret or act on scores in terms of decision-making.

To better understand how surgeons interpret and act on the scores of patient-reported measures in the care of individual patients, we performed an online scenario-based experiment in which scores of various patient-reported measures were randomized into fictional scenarios and asked: 1) What factors are associated with surgeon concern regarding the scores of patient-reported measures in a fictional patient scenario? 2) What factors are associated with surgeon likelihood to offer discretionary surgery? 3) What factors are associated with surgeon motivation to

discuss mental health or social circumstances? 4) What factors are associated with surgeon likelihood to refer for mental and/or social health services?

## Methods

### Study design and setting

Upper limb specialists from the Science of Variation Group (SOVG) were invited via email between May 5th 2023 and June 1st 2023 to participate in a series of experiments addressing the interpretation of PROMs in the care of individual patients (S1 Appendix). In the first stage, a structured questionnaire was used to assess the use of PROMs quantifying capability and mental health in the daily practice of surgeons and to explore the impact, or potential impact, of these scores on care strategies (S2 Appendix).

In the second stage of this pilot work, surgeons reviewed four fictious patient scenarios with PROMs of capability and mental health with varied PROM score distribution (Gaussian and non-Gaussian) for capability and mental health (S2 Appendix). Surgeons indicated which score values concerned them and explained their reasoning, including patient characteristics and perceived barriers to actions. We found that PROMs were infrequently used in practice (40% collected capability scores, 14% mental health scores), and most surgeons did not rely on threshold scores to guide decisions. When asked to interpret scores without guidance, surgeons tended to use statistical benchmarks (e.g., ≥2 SD or top quartile). Some considered high discomfort or incapability a reason to delay surgery and address mental health first, while others saw it as an indication to operate. Factors such as legal disputes also influenced interpretation. Barriers to addressing psychosocial issues included patient unawareness of mental health's role and reluctance to engage in mental or social health support. We assessed the interrater reliability of surgeon rated threshold scores of concern for the 3 ratings ranging between 0 and 100 with normal (2 cases) and non-normal (1 case) distribution. One rating was on a different scale (3 to 15) and could not be included in the reliability analysis. We calculated a two-way mixed effect intraclass correlation coefficient as a measure of variation between specialists and found poor agreement (intraclass correlation coefficient of 0.21; 95% CI: 0.054 to 0.92; S1 Appendix).

In a third stage, building on the first two experiments, all upper limb specialists from the SOVG were invited again to review 10 fictitious scenarios of patients with osteoarthritis (OA) in the upper extremity with randomized elements. We randomized the following elements: age (45, 61 or 83), gender (woman or man), type of pathophysiology (including location of OA and radiographic severity), and scores of fictional patient-reported measures capturing levels of comfort and capability, mental health, and social health. of the trapeziometacarpal (TMC), wrist, and elbow joints were chosen to allow measurement of disease severity as classified from radiographs among different diagnoses. We categorized the pathophysiologic severity at the TMC joint using the Sodha grade, at the wrist using the Scapholunate advanced collapse (SLAC) stage, and at the elbow using the Broberg-Morrey (B-M) grade [9–11]. We classified Sodha grade 1/ SLAC stage 2/ B-M grade 2 as 'mild', Sodha grade 2/ SLAC stage 3/ B-M grade 3 as 'moderate', and Sodha grade 3/ SLAC stage 4/ B-M grade 4 as 'severe'. We also randomized various values of fictious PROMs resembling levels of pain intensity and incapability, unhelpful thoughts and feelings of distress (mental health), and social deprivation index (social health). In each scenario, surgeons were informed that all measures (except pain intensity) were scaled from 0 (best possible score) to 100 (worst possible score), in which 50 represented the general population mean, with 10 points higher or lower denoting a standard deviation. Pain intensity scores were presented on a scale from 0 (no pain) to 10 (worst pain imaginable). We decided to randomize the pain intensity and incapability scores together to avoid discrepancy in the scenarios (e.g., high pain intensity with low incapability). The scores were randomized to represent "low", "average", or "high" values, in which average scores were near the mean, and low or high scores were between 1.5 to 2.5 standard deviations from the mean.

Participants rated (on a 0–100 scale) their level of concern for the patient, likelihood to offer discretionary surgery, motivation to discuss mental and social health, and likelihood to refer to related services (e.g., a social worker or behavioral

therapist), with higher scores indicating greater concern, likelihood, or motivation. The surveys were presented to surgeons using Survey Monkey (Palo Alto, CA, USA).

## Participants

On July 14th, 2023, an email invitation was sent to all upper limb specialist members of the SOVG, an international collaboration of musculoskeletal surgeons that studies sources of variation in care. Participating members receive group authorship or acknowledgment, and there were no financial incentives for their participation. To limit bias, the study participants were not briefed on the study's purpose. Completion of the surveys implied consent. The study protocol, including this approach to consent, was reviewed and approved by the institutional ethics board. Following the initial invitation, non-responders received weekly reminders for three weeks. The study was closed on August 8th, 2023. Among approximately 150 upper limb surgeons that completed at least one survey in the last year, a total of 67 surgeons (45%) participated and completed 89% (598 of 670) of all presented scenarios and 87% of the surgeons [58 of 67] completed all 10 scenarios. Of the remaining surgeons, six completed one scenario, one completed two scenarios and one completed five scenarios. As each surgeon received a random selection of scenarios, the missing data is unlikely to be systematically related to specific case types, minimizing the risk of bias. Most surgeons were men (90%), practice in the United States (61%), supervise trainees (85%), and are specialized in hand and wrist (64%) (Table 1).

## Statistical analysis

We used descriptive statistics to present surgeon demographics. Categorical variables were presented as number with percentage and continuous variables were presented as mean with standard deviation or median with interquartile range dependent on data distribution.

We sought whether surgeon interpretation of scores derived from PROMs capturing comfort and capability, mental health and social health were associated with 1) surgeon concern, 2) likelihood to offer surgery, 3) motivation to discuss mental and social health, and 4) likelihood to refer for mental and social health services. We used multilevel mixed-effects linear regression models to measure associations between these outcomes and randomized patient characteristics

**Table 1. Characteristics of 67 surgeon participants.**

| Variables | Percentage (number) |
|---|---|
| **Men** | 90% (60) |
| **Location of practice** | |
| US | 61% (41) |
| Europe | 19% (13) |
| Other | 19% (13) |
| **Years of experience** | |
| 0 - 5 | 31% (21) |
| 6 - 10 | 24% (16) |
| 11 - 20 | 28% (19) |
| 21 - 30 | 16% (11) |
| **Supervising trainees** | 85% (57) |
| **Subspecialty** | |
| Hand and wrist | 64% (43) |
| Shoulder and elbow | 18% (12) |
| Orthopedic trauma | 13% (9) |
| Other | 4% (3) |

(different scores for various PROMs, age, gender, affected joint, and OA severity), while accounting for nesting by surgeon (non-independence as each surgeon rated multiple cases) [12]. We reported the standardized regression coefficient (RC), 95% confidence interval (95%CI), and corresponding *P*-values. All *P*-values below 0.05 were considered statistically significant. Quantitative data was analyzed using StataMP (Stata 14.0, StataCorp, College Station, TX).

An a priori sample size calculation was performed using G*Power (version 3.1.9.7). Considering the complexity of accurately specifying the parameters and nesting structure of a multilevel mixed-effects model (e.g., intraclass correlation, variability between raters), we opted to perform a sample size calculation for a simpler multivariable linear regression model [13]. This approach allowed us to estimate the required sample size based on anticipated effect size when all included variables account for 15% of the variation in the assessed outcome ($f^2 = 0.18$), desired statistical power (0.80), and significance level (alpha set at 0.05) without relying on potentially unreliable assumptions about nesting by surgeon. This calculation determined that 89 unique observations were required. Given that the achieved sample size of 598 observations (58 participants completed all 10 scenarios, 2 participants completed 5 scenarios, 1 participant completed 2 scenarios, and 6 participants completed 1 scenario) exceeded the a priori estimate by a large margin, it was considered sufficient.

## Results

### Surgeon concern regarding the patient

Accounting for potential confounders such as the patient's age and gender as well as nesting by surgeon, we found that greater surgeon concern regarding an individual based on pathophysiologic severity and scores of fictional PROMs was associated moderately with wrist OA (RC = 4.3, 95%CI = 1.3 to 7.2) and elbow OA (RC = 3.5, 95%CI = 0.58 to 6.4) compared to TMC OA, moderately to strongly with higher scores of distress and unhelpful thoughts (average: RC = 4.5, 95%CI 1.5 to 7.4; and high: RC = 12, 95%CI = 8.9 to 15), moderately with high social deprivation index scores (RC = 5.1, 95%CI = 2.1 to 8.1), and strongly with higher scores of discomfort and incapability (average: RC = 8.4, 95%CI = 5.4 to 12; and high: RC = 17, 95%CI = 14–20) (Tables 2 and 3). Lower surgeon concern regarding an individual was associated moderately with 61 years old age (RC = −3.5, 95%CI = −6.5 to −0.57) compared to 45 years old.

### Likelihood to offer discretionary surgery

Accounting for potential confounders such as the patient's age and gender as well as nesting by surgeon, we found that greater likelihood to offer surgery based on pathophysiologic severity and scores of fictional PROMs was associated strongly with higher radiographic OA severity (moderate: RC = 10, 95%CI = 6.7 to 14; and severe: RC = 15, 95%CI = 11–18), moderately with wrist OA (RC = 5.9, 95%CI = 2.2 to 9.6) and elbow OA (RC = 4.9, 95%CI = 1.3 to 8.5) relative to TMC OA, moderately with average social deprivation index scores (RC = 3.8, 95%CI = 0.081 to 7.5) compared to low social deprivation, and moderately to strongly with higher scores of discomfort and incapability (average: RC = 7.1, 95%CI = 3.2 to 11; and high: RC = 13, 95%CI = 9.0 to 16) (Tables 2 and 3). A lower likelihood to offer surgery was moderately associated with high scores of distress and unhelpful thinking (RC = −8.3, 95%CI = −12 to −4.6) compared to low scores.

### Motivation to discuss mental health and/or social circumstances

Accounting for potential confounders such as the patient's age and gender as well as nesting by surgeon, we found that greater motivation to discuss mental health and social circumstances based on pathophysiologic severity and scores of fictional PROMs was associated moderately to strongly with higher scores of distress and unhelpful thoughts (average: RC = 7.1, 95%CI = 3.8 to 10; and high: RC = 17, 95%CI = 14–21), moderately with high social deprivation scores (RC = 6.1, 95%CI = 2.7 to 9.4), and moderately to strongly with higher scores of discomfort and incapability (average: RC = 5.0, 95%CI = 1.5 to 8.4; and high: RC = 13, 95%CI = 9.6 to 16) (Tables 2 and 3). Lower motivation to discuss mental health and social circumstances was associated moderately with elbow OA (RC = −4.3, 95%CI = −7.6 to −1.1) compared to TMC OA.

## Likelihood to refer for mental and/or social health services

Accounting for potential confounders such as the patient's age and gender as well as nesting by surgeon, we found that greater likelihood to refer for mental and/or social health services based on pathophysiologic severity and scores of fictional PROMs was associated moderately to substantially with higher scores of distress and unhelpful thoughts (average: RC = 6.8, 95%CI = 3.4 to 10; and high: RC = 21, 95%CI = 17–24), moderately with high deprivation index scores (RC = 6.9, 95%CI = 3.5 to 10), and strongly with high scores of discomfort and incapability (RC = 12, 95%CI = 9.1 to 16) compared to low scores (Tables 2 and 3). Lower likelihood to refer was associated moderately with elbow AO (RC = −3.4, 95%CI = −6.7 to −0.051) and worse radiographic severity (moderate: RC = −6.3, 95%CI = −9.8 to −2.8; and severe: RC = −6.0 95%CI = −9.4 to −2.5) (Tables 2 and 3).

**Table 2. Summary of surgeon responses to patient scenarios.**

| | Level of concern | Likelihood to offer Surgery | Motivation to discuss mental and/or social health | Likelihood to refer for mental/ social healthcare |
|---|---|---|---|---|
| | Mean (SD) | Mean (SD) | Mean (SD) | Mean (SD) |
| | 52 (21) | 42 (24) | 49 (24) | 49 (25) |
| **Age** | | | | |
| 45-year-old | 54 (22) | 43 (24) | 49 (24) | 47 (25) |
| 61-year-old | 52 (20) | 42 (22) | 49 (22) | 50 (24) |
| 83-year-old | 52 (21) | 41 (25) | 49 (24) | 51 (25) |
| **Gender** | | | | |
| Men | 52 (22) | 43 (24) | 48 (24) | 48 (24) |
| Women | 54 (20) | 41 (23) | 49 (23) | 49 (23) |
| **Radiographic Severity** | | | | |
| Mild | 52 (21) | 33 (22) | 51 (25) | 51 (25) |
| Moderate | 51 (21) | 44 (23) | 46 (24) | 46 (24) |
| Severe | 55 (21) | 48 (24) | 49 (22) | 49 (22) |
| **Joint** | | | | |
| Trapeziometacarpal | 50 (22) | 39 (25) | 50 (25) | 50 (25) |
| Wrist | 54 (20) | 44 (23) | 49 (24) | 49 (24) |
| Elbow | 54 (21) | 43 (23) | 47 (21) | 47 (21) |
| **Level of distress and unhelpful thinking** | | | | |
| Low | 47 (21) | 45 (25) | 42 (22) | 42 (22) |
| Average | 51 (22) | 43 (23) | 46 (22) | 46 (22) |
| High | 59 (19) | 38 (22) | 57 (23) | 57 (23) |
| **Level of discomfort and incapability** | | | | |
| Low | 43 (21) | 35 (23) | 42 (24) | 42 (24) |
| Average | 51 (19) | 43 (20) | 46 (21) | 46 (21) |
| High | 62 (19) | 47 (25) | 56 (23) | 56 (23) |
| **Deprivation index score** | | | | |
| Low | 51 (21) | 40 (24) | 47 (23) | 47 (23) |
| Average | 53 (20) | 45 (24) | 48 (23) | 48 (23) |
| High | 55 (22) | 40 (23) | 51 (25) | 51 (25) |

All ratings on a 0–100 scale with higher numbers indicating more of the quality being rated.

**Table 3. Mixed multi level linear regression analysis of patient factors associated with surgeons decision making.**

| | Surgeon concern regarding the patient | | | Likelihood to offer discretionary surgery | | | Motivation to discuss mental health and/or social circumstances | | | Likelihood to refer to mental and social health services | | |
|---|---|---|---|---|---|---|---|---|---|---|---|---|
| | RC (95% CI)* | Standard Error | P-value | RC (95% CI)* | Standard Error | P-value | RC (95% CI)* | Standard Error | P-value | RC (95% CI)* | Standard Error | P-value |
| **Age** | | | | | | | | | | | | |
| 45-year | Reference value | | | Reference value | | | Reference value | | | Reference value | | |
| 61-year | −3.5 (−6.5 to −0.57) | 1.2 | 0.19 | −3.0 (−6.7 to 0.74) | 1.9 | 0.12 | −0.60 (−3.9 to 2.7) | 1.7 | 0.64 | 1.3 (−2.1 to 4.7) | 1.7 | 0.46 |
| 83-year | −2.8 (−5.6 to 0.056) | 1.5 | 0.055 | −2.1 (−5.8 to 1.5) | 1.9 | 0.25 | −1.6 (−4.9 to 1.6) | 1.7 | 0.33 | −1.2 (−2.2 to 4.6) | 1.7 | 0.49 |
| **Gender** | | | | | | | | | | | | |
| Male | Reference value | | | Reference value | | | Reference value | | | Reference value | | |
| Female | 2.3 (−0.23 to 4.7) | 1.2 | 0.052 | −0.85 (−3.9 to 2.1) | 1.5 | 0.58 | −0.64 (−3.3 to 2.0) | 1.4 | 0.64 | −0.3 (−3.1 to 2.5) | 1.4 | 0.83 |
| **Radiographic Severity** | | | | | | | | | | | | |
| Mild | Reference value | | | Reference value | | | Reference value | | | Reference value | | |
| Moderate | 0.78 (−2.3 to 3.8) | 1.5 | 0.61 | 10 (6.7 to 14) | 1.9 | **<0.001** | −2.2 (−5.6 to 1.2) | 1.7 | 0.21 | −6.3 (−9.8 to −2.8) | 1.8 | **<0.001** |
| Severe | 2.7 (−0.31 to 5.6) | 1.5 | 0.079 | 15 (11 to 18) | 1.9 | **<0.001** | −2.7 (−6.1 to 0.61) | 1.7 | 0.11 | −6.0 (−9.4 to −2.5) | 1.8 | **0.001** |
| **Joint** | | | | | | | | | | | | |
| Trapeziometacarpal | Reference value | | | Reference value | | | Reference value | | | Reference value | | |
| Wrist | 4.3 (1.3 to 7.2) | 1.5 | **0.004** | 5.9 (2.2 to 9.6) | 1.9 | **0.002** | −2.2 (−5.5 to 1.1) | 1.7 | 0.19 | −1.2 (−4.6 to 2.2) | 1.7 | 0.494 |
| Elbow | 3.5 (0.58 to 6.4) | 1.5 | **0.018** | 4.9 (1.3 to 8.5) | 1.8 | **0.008** | −4.3 (−7.6 to −1.1) | 1.7 | **0.009** | −3.4 (−6.7 to −0.051) | 1.7 | **0.047** |
| **Level of distress and unhelpful thinking** | | | | | | | | | | | | |
| Low | Reference value | | | Reference value | | | Reference value | | | Reference value | | |
| Average | 4.5 (1.5 to 7.4) | 1.5 | **0.003** | −3.1 (−6.8 to 0.55) | 1.9 | 0.095 | 7.1 (3.8 to 10) | 1.7 | **<0.001** | 6.8 (3.4 to 10) | 1.7 | **<0.001** |
| High | 12 (8.9 to 14.7) | 1.5 | **<0.001** | −8.3 (−12 to −4.6) | 1.9 | **<0.001** | 17 (14 to 21) | 1.7 | **<0.001** | 21 (17 to 24) | 1.7 | **<0.001** |
| **Discomfort** | | | | | | | | | | | | |
| Low | Reference value | | | Reference value | | | Reference value | | | Reference value | | |
| Average | 8.4 (5.4 to 12) | 1.6 | **<0.001** | 7.1 (3.2 to 11) | 1.9 | **<0.001** | 5.0 (1.5 to 8.4) | 1.7 | **0.004** | 3.4 (−0.16 to 6.9) | 1.8 | 0.061 |
| High | 17 (14 to 20) | 1.5 | **<0.001** | 13 (9.0 to 16) | 1.9 | **<0.001** | 13 (9.6 to 16) | 1.7 | **<0.001** | 12 (9.1 to 16) | 1.7 | **<0.001** |
| **Deprivation index** | | | | | | | | | | | | |

*(Continued)*

**Table 3.** (Continued)

| | Surgeon concern regarding the patient | | | Likelihood to offer discretionary surgery | | | Motivation to discuss mental health and/or social circumstances | | | Likelihood to refer to mental and social health services | | |
|---|---|---|---|---|---|---|---|---|---|---|---|---|
| | RC (95% CI)* | Standard Error | P-value | RC (95% CI)* | Standard Error | P-value | RC (95% CI)* | Standard Error | P-value | RC (95% CI)* | Standard Error | P-value |
| Low | Reference value | | | Reference value | | | Reference value | | | Reference value | | |
| Average | 2.3 (−0.69 to 5.2) | 1.5 | 0.13 | 3.8 (0.081 to 7.5) | 1.9 | **0.045** | 1.4 (−1.9 to 4.7) | 1.7 | 0.41 | 1.6 (−1.9 to 5.0) | 1.7 | 0.371 |
| High | 5.1 (2.1 to 8.1) | 1.5 | **0.001** | −0.22 (−3.9 to 3.5) | 1.9 | 0.906 | 6.1 (2.7 to 9.4) | 1.7 | **<0.001** | 6.9 (3.5 to 10) | 1.7 | **<0.001** |

**Bold** indicates statistical significance, P < 0.05. *Regression Coefficient (95% Confidence Interval).

## Discussion

Patient-reported outcome measures quantify the patient's experience of health, such as levels of musculoskeletal discomfort and incapability [14]. The analysis of PROMs in studies comparing tests or treatment strategies or seeking factors associated with such scores is straightforward, but it is relatively unclear how specialists interpret scores of quantified health in the context of individual patient care. In this multi-staged scenario-based study, we found that only 40% collected capability PROMs and 14% collected mental health PROMs. Surgeons interpreted scores as concerning mostly based on standard deviations and interquartile range (2 standard deviations above mean or highest quartile), and there was poor interrater agreement regarding the surgeon-rated threshold of concern (surgeon reaction to PROMs is highly variable). Surgeon concern for patients was primarily associated with higher pain intensity and incapability scores, and to a slightly lesser degree with worse mental and social health scores. These variables also motivated surgeons to discuss mental health and social circumstance with patients and refer them to mental and/or social health services, while worse radiographic OA severity was associated with lower likelihood to refer. Surgeons were more likely to offer surgery in patients with more severe radiographic OA severity and higher pain intensity and incapability scores, and less likely to offer surgery in patients with high scores of distress and unhelpful thoughts.

### Surgeon concern regarding the patient

The observation that surgeon interpretation of PROMs was related to notable deviations from expected (mean) scores, in which surgeons rated scores as concerning above 2 standard deviations (2.5% of the population) or in the top quartile (25% of the population) (S1 Appendix), merits concern. Surgeons may not realize to which extent these interpretations lead to a different understanding of scores related to personal aspects of illness, and these findings direct us to educate surgeons about the influence of data distribution on the interpretation of patient-reported scores.

The observation that surgeon concern regarding patients was associated with greater scores of discomfort and incapability, greater scores of distress and unhelpful thinking, and greater social deprivation index scores suggests that surgeons are able to interpret PROMs in a comprehensive way, conceptualizing them within the biopsychosocial paradigm of human illness. The relative strength of association as judged by the regression coefficients suggest that surgeon concern is most strongly associated with levels of discomfort and incapability, followed by levels of distress and unhelpful thinking. Surgeons seem to recognize that relatively high levels of discomfort and incapability on patient-reported measure scores can be a signal to consider mental health and social circumstances. This might reflect a growing awareness of the notable evidence that a large proportion of the variation in levels of discomfort and incapability is accounted for by mindset (thoughts and feelings), relatively independent of pathophysiological severity [6,15–20].

The observation that TMC OA elicited comparatively lower concern among surgeons compared to wrist OA and elbow OA could be attributed to the near universal prevalence of TMC OA with age (expected senescence) [21], which may coincide with participant anticipation of a notable rate of effective accommodation to which patients could be guided. In contrast, wrist OA and elbow OA are less common and often post-traumatic [22,23].

## Likelihood to offer discretionary surgery

The finding that surgeons were more likely to offer surgery for patients with greater pain intensity and incapability scores, worse radiographic severity, and those with elbow or wrist OA implies that surgeons interpret these factors as indicative of greater potential for benefit by addressing pathophysiology. Surgeons might sense that surgery is best suited for those with advanced pathophysiology and/or greater scores of pain intensity and incapability when there are low levels of psychosocial concern. However, it is important to keep in mind that these findings are based on fictious patient scenarios in which randomized scores may have presented an unlikely or uncommon scenario such as high levels of discomfort and incapability with low levels of distress, unhelpful thoughts, and social deprivation. Given that surgeons may tend to anchor on surgical treatment options there may be a role for a checklist to remind surgeons to consider prioritizing mental and social health when surgery is discretionary (optional) and not time-sensitive (elective) [24]. It is important that surgeons keep in mind that disproportionately greater discomfort can be a signal of important mental and social health factors [6,15–19], and that there is a relatively limited association between radiographic severity in OA and symptom intensity [25,26]. The observation that lower likelihood to offer surgery is associated with an average deprivation index is difficult to interpret. This may reflect unmeasured socioeconomic factors influencing decision-making or access to care. Give the exploratory nature of this study, we advise caution in interpreting this finding.

## Motivation to discuss mental health and/or social circumstances and likelihood to refer to mental and social health services

The findings that surgeons exhibit greater motivation to discuss mental and social health when a patient has relatively greater scores of pain intensity and incapability, distress and unhelpful thinking, and social deprivation reiterates that PROMs serve as cues for specialists to address these aspects of patient well-being. Similarly, the observation that surgeons are more inclined to refer a patient for mental and social health support based on greater scores of pain intensity and incapability, distress and unhelpful thinking, and social deprivation, suggests that PROMs could facilitate comprehensive care strategies. This is intriguing, particularly in comparison to previous SOVG studies [27,28], which found that surgeons were likely to recognize mental health issues but remained neutral regarding referrals due to perceived barriers. This might reflect the potential utility of incorporating PROMs into routine care in order to help nudge surgeon awareness of certain aspects of the patient's health that might be prioritized. Especially considering previous evidence has shown that priming surgeons to be more attuned for opportunities to address mental health can increase the identification of feelings of distress and unhelpful thoughts [29]. Integration of PROMs to attune surgeons may further enhance identification and addressment of mental health opportunities. Recent studies underscore that surgeons express willingness to address psychological factors, but also report practical barriers including 1) perceived time constraints, 2) discomfort with discussing mental and social health, 3) perceived societal stigma hindering effective referrals for mental and social health, and 4) perceived lack of access to resources to provide mental and social health care [27,30]. The preparatory work for this study identified similar barriers (S1 Appendix).

The observation that worse radiographic OA severity and involvement of the elbow decreased surgeon likelihood to refer again suggests some variations in care tendencies related to pathophysiology. Surgeons may associate excessive pain and distress with severe arthritis, influencing their decision-making regarding referrals. It is also possible that surgeons feel more confident or effective in treating pathophysiology, and perhaps think that it could improve a person's

mental health as well. Yet, addressing mental and social health needs may benefit someone regardless of their OA severity.

## Limitations

This study can be considered in light of several limitations. First, the participants in this study were predominantly men working in Europe or the United States and 85% were supervising trainees indicating that most of the surgeons work in academic environments. Therefore, the measured rates of categorical variables and means and medians of continuous variables may not be reproduced in other samples of specialists. However, the statistical relationships between variables may be reproducible in any sample with adequate variation in opinions as can be seen in the wide standard deviations in our response variables. Prior SOVG studies confirm a diversity of opinions among participants [29,31,32]. Second, some of the observed variation may relate to the wording and interpretation of the scenarios and alternative wordings could be explored in future studies. Third, the use of fictional patient scenarios does not capture the intricacies and nuances of actual patient encounters. However, the relative simplicity of the printed scenarios might help isolate potentially important factors and determine if more sophisticated studies observing the opinions and behaviors of clinicians with actual patients are worthwhile. Moreover, because the patient characteristics were randomly generated, some scenarios may have presented unlikely or uncommon combinations, such as high levels of discomfort and incapability alongside low levels of distress, unhelpful thoughts, and social deprivation. Fourth, we focused on a single specialty visit as claims data suggest that most patients see a specialist a single time [7]. This precluded the use of benchmarks such as minimal important differences over time. Fifth, our scenarios only presented cases of OA in the upper extremity. Although not directly tested, we hypothesize that the interpretation of PROMs and the influence on treatment recommendation for other non-emergent musculoskeletal conditions would be comparable. We chose to focus on OA because radiographs can be used as a representation of the severity of the pathophysiology. Some readers might consider this an inadequate measure of pathophysiology, but imaging is the best currently available representation of OA severity. Sixth, a small number of surgeons exited the survey before rating all the scenarios. Since each scenario is randomized and unique, this should not alter the findings appreciably. Seventh, the participants likely gleaned the purpose of the study even though it was not explicitly stated, which could result in a Hawthorne effect [33], although it's difficult to know the degree to which research participation effects lead to misleading results.

## Conclusion

The observation that surgeon care strategies for individual patients with upper extremity OA vary according to the scores on PROMs that quantify levels of discomfort and capability, mental health, and social health suggests that surgeons use such measures to inform comprehensive, whole-person care strategies. These findings may apply to musculoskeletal illness more broadly. On the other hand, most surgeons did not collect PROMs in their own practice, displayed low agreement about thresholds of concern, and interpreted scores differently based on data distribution, indicating the need to develop best practices for the use of PROMs in the care of individual patients. A next step could be to assess patient and clinician behavior in a setting of routine collection of PROMs in daily practice. Future investigations could explore the effectiveness of biopsychosocial treatment strategies tailored to scores of PROMs quantifying the level of comfort, capability, mental health and social health.

## Supporting information

**S1 Appendix. Results of preparatory experiments.**
(DOCX)

**S2 Appendix. Surveys corresponding to the preparatory and main experiments.**
(DOCX)

**S3 Appendix. Underlying data retrieved from SurveyMonkey for analyses.**
(XLSX)

## Acknowledgments

We thank the following SOVG members for participating in this study:

Edward Harvey, Michael W Grafe, Betsy M. Nolan, Andreas Platz, Louis Christopher Grandizio, Todd Bafus, Sebastian von Unger, Bernard F Hearon, Richard S. Page, Karel Chivers, David Napoli, David Ruch, Gerald A. Kraan, Eitan Melamed, James Popp, Patrick W. Owens, Michael Cohen, Ngozi M. Akabudike, Craig Rodner, Niels W.L. Schep, Minoo Patel, Frank IJpma, Gregory DeSilva, Richard S. Gilbert, Nathan A Hoekzema, Steve Kronlage, Ralf Walbeehm, Sebastiaan Souer, Michael Jason Palmer, Carlos Henrique Fernandes, Adam Shafritz, John M. Stephenson, Ramon de Bedout, Jason D. Tavakolian, M. van der Pluijm, Naquira Escobar Luis Felipe, Marc J. Richard, Stephen A. Kennedy, Bradley A. Palmer, Daniel London, Kevin Rumball, Anne J.H. Vochteloo, Ryan P. Calfee, Julie Balch Samora, Maurizio Calcagni, Juan M Patiño, Taizoon Baxamusa, H Brent Bamberger, German Ricardo Hernandez, Constanza L. Moreno-Serrano, Todd Siff, Michel van den Bekerom, Roger van Riet, Steven L. Henry, Thomas Apard, Michael Nancollas, Kristin Karim, James F. Nappi, Chris Casstevens, Jeffrey Wint, Nata Parnes, Eric P. Hofmeister, Robert R.L. Gray, Frank W. Bloemers, Vasileios S. Nikolaou, Pradeep Choudhari, Theodoros Tosounidis, Andrew John Powell, Diederik O. Verbeek, Robert E. Van Demark Jr, Amanda I. Gonzalez, Philipp Muhl, Percy V. van Eerten, Lisa Nash, Nathan A. Hoekzema, Rejith Valsalan Mannambeth, William B. Ericson Jr., Richard A. Schaefer, Jacob Gire, Ippokratis Pountos, William Dias Belangero, Yoram Weil, Thomas Apard, Emilia Stojkovska Pemovska, Thomas Mittlmeier, Dan Polatsch, Eric Raven, Hans Goost, Daniel Haverkamp, Tim Chesser, Tim Schepers, Peter Schandelmaier, Steven J. Morgan, Andrew Neviaser, Ladislav Mica, Prosper Benhaim, Sebastian Farr, Yohan Jang, George Pianka, Erik T. Walbeehm, Iain McGraw, Richard Jenkinson, Koroush Kabir, John P. Evans, Nina Lightdale-Miric, Gregory J. Della Rocca, Clay Spitler, C. Liam Dwyer, Kevin Rumball, Ramon de Bedout, Julie Balch Samora, Naquira Escobar Luis Felipe, Michael Prayson, Richard Wallensten, George Babis, Vincenzo Giordano, Lob Guenter, Christos Garnavos, Ante Prkic, Thomas De Coster, Lisa Taitsman, Jan Debeij, Tomo Havliček, Daniel C. Wascher, L.W. van der Plaat., Christiaan. J.A. van Bergen, Efstathios G. Ballas, Ellen Satteson, James F. Nappi, John M. Erickson, Quell M., Juan C. Cagnone, Minos Tyllianakis, Thierry Begue, Poonam Harish, Alexander Marcus, Antonio Barquet, Daniel B. Whelan, Jose Eduardo Grandi Ribeiro Filho, Richard Buckley, Kathryn Doughty, Edward K. Rodriguez, Sean T. Campbell, Linda Vallejo, Giselly Veríssimo de Miranda, Jorge Rubio, Miguel Pirela Cruz, Lawrence S. Halperin, Julie Adams, Katsunori Suzuki, Sanjeev Kakar, Russell Shatford, Lisa Cannada, Lodewijk M.S.J. Poelhekke, Cecile van Laarhoven, Jesus Moreta, Frede Frihagen, Elizabeth Martin, Marc Swiontkowski, Mathieu M.E. Wijffels

## Author contributions

**Conceptualization:** Rick S. Looman, Niels Brinkman.

**Data curation:** Rick S. Looman, Sina Ramtin.

**Formal analysis:** Rick S. Looman, Niels Brinkman.

**Investigation:** Rick S. Looman, Koen D. Oude Nijhuis, Niels Brinkman.

**Methodology:** Rick S. Looman, David Ring, Niels Brinkman.

**Project administration:** Rick S. Looman, Sina Ramtin.

**Resources:** David Ring.

**Supervision:** David Ring, Daniel A. London, Ryan Calfee, Niels Brinkman.

**Writing – original draft:** Rick S. Looman, David Ring.

**Writing – review & editing:** Koen D. Oude Nijhuis, David Ring, Daniel A. London, Ryan Calfee, Niels Brinkman.

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
