## [Decision Letter · Decision Letter 0]

26 Jul 2025

PONE-D-25-05662Factors associated with surgeon interpretation of PROM values and their prioritization of treatment based on those scoresPLOS ONE

Dear Dr. Ring,

Thank you for submitting your manuscript to PLOS ONE. After careful consideration, we feel that it has merit but does not fully meet PLOS ONE’s publication criteria as it currently stands. Therefore, we invite you to submit a revised version of the manuscript that addresses the points raised during the review process.

We look forward to receiving your revised manuscript.

Kind regards,

Xindie Zhou

Academic Editor

PLOS ONE

Journal Requirements:

2. Thank you for stating the following in the Competing Interests/Financial Disclosure section: I have read the journal's policy and the authors of this manuscript have the following competing interests: David Ring reports other from Skeletal Dynamics, personal fees from Deputy Editor for Clinical Orthopaedics and Related Research, personal fees from Universities and Hospitals, personal fees from Lawyers, personal fees from Health Services and Resource Administration and Department of Justice, personal fees from Premier Healthcare Solutions, personal fees from Wolters Kluwer Health, grants from National Institutes for Health, personal fees from Everus, other fromMyMedicalHub, outside the submitted work.

Ryan Calfee receives grants from the NIMH, serves as Deputy Editor in Chief Journal of Hand Surgery and servers as Research Director American Society for Surgery of the Hand.

All other authors have no conflicts of interest to declare.

We note that one or more of the authors have an affiliation to the commercial funders of this research study: Premier Healthcare Solutions, Wolters Kluwer Health, Everus

3. One of the noted authors is a group or consortium science of variation group. In addition to naming the author group, please list the individual authors and affiliations within this group in the acknowledgments section of your manuscript. Please also indicate clearly a lead author for this group along with a contact email address.

4. In the online submission form, you indicated that the data underlying the results presented in the study were collected via SurveyMonkey and are stored securely by the authors. Due to privacy and ethical considerations, the data are not publicly available. However, they can be provided upon reasonable request to the corresponding author.

5. Please ensure that you include a title page within your main document. You should list all authors and all affiliations as per our author instructions and clearly indicate the corresponding author.

6. Please amend either the abstract on the online submission form (via Edit Submission) or the abstract in the manuscript so that they are identical.

Reviewers' comments:

Reviewer's Responses to Questions

**Comments to the Author**

1. Is the manuscript technically sound, and do the data support the conclusions?

Reviewer #1: Yes

Reviewer #2: Partly

2. Has the statistical analysis been performed appropriately and rigorously? 

Reviewer #1: Yes

Reviewer #2: Yes

3. Have the authors made all data underlying the findings in their manuscript fully available?

Reviewer #1: Yes

Reviewer #2: Yes

4. Is the manuscript presented in an intelligible fashion and written in standard English?

Reviewer #1: Yes

Reviewer #2: Yes

5. Review Comments to the Author

Reviewer #1: This study explores how surgeons explain patient-reported outcome measures (PROMs) in clinical practice, and it is innovative in its topic selection. The research questions have significant clinical relevance, and the research methods are generally reasonable, but there are still several issues that require attention.

1.Lines 1-2: The title accurately reflects the study content, though it could be more concise. Consider: "Surgeon interpretation and clinical application of patient-reported outcome measures in upper extremity osteoarthritis."

2.Lines 70-94: The description of preliminary work is helpful for understanding the study development, but this section is somewhat lengthy and could be condensed.

3.Please revise the reference citation format in the manuscript.

4.The situation where full names and abbreviations such as PROMs or PROM, SOVG, etc. are used interchangeably needs to be checked throughout this manuscript.

5.Line 242: The interrater reliability finding (ICC = 0.21) deserves more prominent discussion as it suggests significant variability in PROM interpretation.

6.Lines 135�Although 67 surgeons completed 89% of the scenarios, there was a lack of analysis regarding the patterns of the incomplete scenarios and potential biases.

7.Among the 67 participants in the manuscript, 90% were male. I believe the author should consider the impact of this situation on the generalizability of the results.

8.It is suggested that the author should consider including a more diverse sample of surgeons in future research, including doctors of different genders and from different practice environments, in order to enhance the external validity of the research results.

9.The manuscript mentions that the implicit form of consent, "completion of the investigation indicates consent", poses hidden ethical risks in ethics.

Reviewer #2: The manuscript addresses an important and underexplored area; the topic is timely and highly relevant in the context of increasing emphasis on patient-centered care and shared decision-making. That said, there are several areas where the clarity, scientific precision, and logical flow of the manuscript, particularly in the results and conclusion sections, could be improved.

Line 24: The sentence is grammatically incorrect and unclear. It seems to be two thoughts merged into one.

Line 16: Typo: “associate” should be associated.

Line 64–70: Clarify timeline: Were stages sequential? Were participants the same?

Line 90-93: State clearly why the interrater reliability was assessed. What did poor ICC mean for the study design?

Line 119–124: Consider grouping the outcome variables with consistent phrasing to avoid redundancy.

Line 132: Clarify why participants were not informed of the study’s purpose was this to reduce bias? Add justification. Mention the IRB/ethics board or waiver of consent if applicable.

Line 157: It would be clearer to state “randomized patient characteristics were included as covariates.

Line 164–165: Consider citing a methodological reference that supports the practice of approximating multilevel power via simpler models.

Line 172: “Sample size used in this study seems to be sufficient” is too informal and speculative.

Line 203: “Average social deprivation” use more precise phrasing.

Line 243–244: Add specific β values or effect size examples for clarity and scientific weight.

Line 285–287: Important point about improbable score combinations. Flag this as a limitation.

Line 288: The checklist suggestion is excellent; provide an example or cite prior use in other specialties.

Line 294: “Average deprivation index” interpretation needs elaboration

Line 330–331: The phrase “rates and ratings” is vague. consider rephrasing it

6. PLOS authors have the option to publish the peer review history of their article (what does this mean? ). If published, this will include your full peer review and any attached files.

**Do you want your identity to be public for this peer review?** For information about this choice, including consent withdrawal, please see our Privacy Policy .

Reviewer #1: No

Reviewer #2: No

---

## [Author Response · Author response to Decision Letter 1]

19 Aug 2025

Dear Editor,

We appreciate the efforts of the editors and reviewers to improve our manuscript. We have addressed the comments in a revision as follows:

Reviewer 1

1. Lines 1-2: The title accurately reflects the study content, though it could be more concise. Consider: "Surgeon interpretation and clinical application of patient-reported outcome measures in upper extremity osteoarthritis."

Response: “Clinical application” may imply real-world implementation or outcomes, which our experimental vignette-based design does not address.

Changes: Changed to: "Surgeon interpretation of patient-reported outcome measures in upper extremity osteoarthritis".

2. Lines 70-94: The description of preliminary work is helpful for understanding the study development, but this section is somewhat lengthy and could be condensed.

Response: OK.

Changes: Section condensed.

3. Please revise the reference citation format in the manuscript.

Response/Changes: Citations reformatted according to PLOS ONE current guidelines.

4. The situation where full names and abbreviations such as PROMs or PROM, SOVG, etc. are used interchangeably needs to be checked throughout this manuscript.

Response: OK.

Changes: We have reviewed the manuscript and revised all instances of full names and abbreviations to ensure consistent and appropriate use. We now consistently use “PROMs” and “SOVG” after the first full mention.

5. Line 242: The interrater reliability finding (ICC = 0.21) deserves more prominent discussion as it suggests significant variability in PROM interpretation.

Response: OK.

Changes: Added more discussion of the poor interrater reliability.

6. Lines 135�Although 67 surgeons completed 89% of the scenarios, there was a lack of analysis regarding the patterns of the incomplete scenarios and potential biases.

Response: We did not perform an analysis of surgeon-related factors and their association with incomplete scenarios. Given the randomization of patient factors and the relatively high completion rate (89%), we considered the likelihood of systematic bias to be low. However, we acknowledge that such an analysis could be of interest and can be explored in future work if deemed relevant.

Changes: We now report that the remaining surgeons completed 1 (n=6), 2 (n=1), or 5 (n=1) scenarios. As each surgeon received a random selection of scenarios, the missing data is unlikely to be systematically related to specific case types, minimizing the risk of bias.

7. Among the 67 participants in the manuscript, 90% were male. I believe the author should consider the impact of this situation on the generalizability of the results.

Response: We agree and addressed this in the limitations. What makes the experiment work is the variation in ratings, which is sufficient. But the rates do not apply outside this sample. We welcome all participants and have done a great deal of recruiting.

Changes: No changes were made to the manuscript text, as this limitation was already addressed in the discussion section.

8. It is suggested that the author should consider including a more diverse sample of surgeons in future research, including doctors of different genders and from different practice environments, in order to enhance the external validity of the research results.

Response/Changes: See response to comment 7 of reviewer 1.

9.The manuscript mentions that the implicit form of consent, "completion of the investigation indicates consent", poses hidden ethical risks in ethics.

Response: Our IRB disagrees with you. The effort of voluntarily entering the survey and completing it is probably a greater affirmation of consent than signing a form. Also consider that the collaboration is 15 years old and most of these are seasoned participants.

Changes: No changes made.

Reviewer 2

1. Line 24: The sentence is grammatically incorrect and unclear. It seems to be two thoughts merged into one.

Response: OK.

Changed to: Sentence revised.

2. Line 16: Typo: “associate” should be associated

Response/Change: Corrected.

3. Line 64–70: Clarify timeline: Were stages sequential? Were participants the same?

Response: Addressed missing timeline and information.

Change: Added the data collection periods for the first and second stages and clarified in the methods section that the stages were sequential with separate participant samples.

4. Line 90-93: State clearly why the interrater reliability was assessed. What did poor ICC mean for the study design?

Response: The study aim was to measure variability of specialist PROM interpretation.

Change: See response to Reviewer 1, Comment 5.

5. Line 119–124: Consider grouping the outcome variables with consistent phrasing to avoid redundancy.

Response: OK.

Change: To improve clarity and reduce redundancy, we revised the sentence to consistently group the outcome variables using parallel phrasing: “Participants rated (on a 0–100 scale) their level of concern for the patient, likelihood to offer discretionary surgery, motivation to discuss mental and social health, and likelihood to refer to related services (e.g., a social worker or behavioral therapist), with higher scores indicating greater concern, likelihood, or motivation.”

6. Line 132: Clarify why participants were not informed of the study’s purpose was this to reduce bias? Add justification. Mention the IRB/ethics board or waiver of consent if applicable.

Response: Participants in SOVG experiments are not informed of the study purpose to limit bias. The study protocol, including this approach to consent, was reviewed and approved by the institutional ethics board.

Changes: See Response 9 to reviewer 1. Clarified in the method that participant were not informed to reduce bias.

7. Line 157: It would be clearer to state “randomized patient characteristics were included as covariates.

Response: OK.

Changes: Clarified as suggested.

8. Line 164–165: Consider citing a methodological reference that supports the practice of approximating multilevel power via simpler models.

Response: Agreed.

Changes: Added citation supporting the practice of multilevel linear models.

9. Line 172: “Sample size used in this study seems to be sufficient” is too informal and speculative.

Response: Agreed and rephrased the sentence.

Changes: Revised to: “The achieved sample size exceeded the a priori estimate and is therefore considered sufficient.”

10. Line 203: “Average social deprivation” use more precise phrasing.

Response: We believe the current phrasing is appropriate. The term “average” is clearly defined in the Methods section (lines 107 and 115), where we explain that average scores reflect values near the population mean, while low and high scores represent values 1.5 to 2.5 standard deviations below or above the mean.

Changes: No changes made. Open to suggestions.

11. Line 243–244: Add specific β values or effect size examples for clarity and scientific weight.

Response: We follow most Editors preference to not restate the numbers in the Discussion. Instead we summarize and interpret them.

Changes: No changes made pending Editor input that they would prefer we restate the numbers.

12. Line 285–287: Important point about improbable score combinations. Flag this as a limitation.

Response: OK.

Changes: Information added to limitation section.

13. Line 288: The checklist suggestion is excellent; provide an example or cite prior use in other specialties.

Response: OK.

Changes: An example of a checklist is described and cited.

14. Line 294: “Average deprivation index” interpretation needs elaboration

Response: It may reflect unmeasured factors related to socioeconomic status influencing surgeon decision-making, patient preferences, or access to care. However, given the exploratory nature of this study using fictional scenarios, we advise caution in overinterpreting this result and suggest it as an area for future research.

Changes: We added a sentence in the discussion section elaborating on the complexity and possible interpretations.

15. Line 330–331: The phrase “rates and ratings” is vague. consider rephrasing it

Response: Agreed.

Changes: Rewritten to “...the measured rates of categorical variables means and medians of continuous variables....”

Editor comments

Response/changes: Manuscript checked to require PLOS ONE’s style requirements.

2. Thank you for stating the following in the Competing Interests/Financial Disclosure section: I have read the journal's policy and the authors of this manuscript have the following competing interests: David Ring reports other from Skeletal Dynamics, personal fees from Deputy Editor for Clinical Orthopaedics and Related Research, personal fees from Universities and Hospitals, personal fees from Lawyers, personal fees from Health Services and Resource Administration and Department of Justice, personal fees from Premier Healthcare Solutions, personal fees from Wolters Kluwer Health, grants from National Institutes for Health, personal fees from Everus, other fromMyMedicalHub, outside the submitted work.

Ryan Calfee receives grants from the NIMH, serves as Deputy Editor in Chief Journal of Hand Surgery and servers as Research Director American Society for Surgery of the Hand.

All other authors have no conflicts of interest to declare.

We note that one or more of the authors have an affiliation to the commercial funders of this research study: Premier Healthcare Solutions, Wolters Kluwer Health, Everus

Response: The study was not funded. We have moved the updated Funding Statement and Competing Interests Statement into the cover letter as requested.

Changes:

Following information was added:

Funding Statement: This research received no specific grant from any funding agency in the public, commercial, or not-for-profit sectors.

Competing interests statement: [DR] is affiliated with Premier Healthcare Solutions, Wolters Kluwer Health, and Everus. This does not alter our adherence to PLOS ONE policies on sharing data and materials.

3. One of the noted authors is a group or consortium science of variation group. In addition to naming the author group, please list the individual authors and affiliations within this group in the acknowledgments section of your manuscript. Please also indicate clearly a lead author for this group along with a contact email address.

Response: We will add the Science of Variation Group (SOVG) as a group author on the authorship list. There is no “lead author” for the group. The lead authors are named authors. But if you want to designate a lead author, please designate Dr. David Ring. Due to practical limitations, we are unable to provide individual affiliations for each member of the group. Instead, if you need to list an affiliation we propose listing the SOVG as generally affiliated with Dell Medical School at The University of Texas in Austin.

Changes: The SOVG has been added as a group author on the title page.

4. All PLOS journals now require all data underlying the findings described in their manuscript to be freely available to other researchers, either a. In a public repository, b. Within the manuscript itself, or c. Uploaded as supplementary information.This policy applies to all data except where public deposition would breach compliance with the protocol approved by your research ethics board. If your data cannot be made publicly available for ethical or legal reasons (e.g., public availability would compromise patient privacy), please explain your reasons on resubmission and your exemption request will be escalated for approval.

Response/Changes: We have prepared the underlying data with all personal health information removed. These data have been uploaded as Supporting Information.

5. Please ensure that you include a title page within your main document. You should list all authors and all affiliations as per our author instructions and clearly indicate the corresponding author.

Response/changes: We have ensured that the main manuscript file includes a complete title page listing all authors and their affiliations, with the corresponding author clearly indicated. In addition, the Science of Variation Group is listed as a group consortium in accordance with the PLOS ONE author instructions.

6. Please amend either the abstract on the online submission form (via Edit Submission) or the abstract in the manuscript so that they are identical.

Response/changes: We have revised the abstract so that the version in the online submission form and the version in the manuscript are now identical.

7. Please include captions for your Supporting Information files at the end of your manuscript, and update any in-text citations to match accordingly.

Response/changes: We have added captions for all Supporting Information files at the end of the manuscript and updated all in-text citations to match PLOS ONE guidelines.

Response/changes: We have reviewed all publications suggested by the reviewers. Relevant works have been cited where appropriate; works deemed not directly relevant were not cited, in line with the editor’s guidance.

---

## [Decision Letter · Decision Letter 1]

28 Aug 2025

Surgeon interpretation of patient-reported outcome measures in upper extremity osteoarthritis

PONE-D-25-05662R1

Dear Dr. Ring,

We’re pleased to inform you that your manuscript has been judged scientifically suitable for publication and will be formally accepted for publication once it meets all outstanding technical requirements.

Kind regards,

Xindie Zhou

Academic Editor

PLOS ONE

Additional Editor Comments (optional):

Reviewer #1:

Reviewer #2:

Reviewers' comments:

Reviewer's Responses to Questions

**Comments to the Author**

1. If the authors have adequately addressed your comments raised in a previous round of review and you feel that this manuscript is now acceptable for publication, you may indicate that here to bypass the “Comments to the Author” section, enter your conflict of interest statement in the “Confidential to Editor” section, and submit your "Accept" recommendation.

Reviewer #1: All comments have been addressed

Reviewer #2: All comments have been addressed

2. Is the manuscript technically sound, and do the data support the conclusions?

Reviewer #1: Partly

Reviewer #2: Yes

3. Has the statistical analysis been performed appropriately and rigorously? 

Reviewer #1: Yes

Reviewer #2: Yes

4. Have the authors made all data underlying the findings in their manuscript fully available?

Reviewer #1: Yes

Reviewer #2: Yes

5. Is the manuscript presented in an intelligible fashion and written in standard English?

Reviewer #1: Yes

Reviewer #2: Yes

6. Review Comments to the Author

Reviewer #1: (No Response)

Reviewer #2: (No Response)

7. PLOS authors have the option to publish the peer review history of their article (what does this mean? ). If published, this will include your full peer review and any attached files.

**Do you want your identity to be public for this peer review?** For information about this choice, including consent withdrawal, please see our Privacy Policy .

Reviewer #1: No

Reviewer #2: No

---

## [Editor Report · Acceptance letter]

PONE-D-25-05662R1

PLOS ONE

Dear Dr. Ring,

I'm pleased to inform you that your manuscript has been deemed suitable for publication in PLOS ONE. Congratulations! Your manuscript is now being handed over to our production team.

Kind regards,

on behalf of

Dr. Xindie Zhou

Academic Editor

PLOS ONE